# Spatial Patterns of Emergency Dental Care Utilization: Assessing the Influence of Social Vulnerability and Dental Provider Availability

**DOI:** 10.3390/healthcare12212191

**Published:** 2024-11-04

**Authors:** Darrell J. Gaskin, Oshin Kanwar, Rachael McCleary, Anna L. Davis, Darien J. Weatherspoon, Eleanor Fleming, Ali Iftikhar, Khushbu Balsara, Hossein Zare

**Affiliations:** 1Department of Health Policy and Management, Johns Hopkins Bloomberg School of Public Health, 750 E Pratt St., Floor 15, Baltimore, MD 21202, USA; dgaskin1@jhu.edu (D.J.G.); oshinkanwar93@gmail.com (O.K.); rmcclear@jhu.edu (R.M.); adavis@gwc.com (A.L.D.); aiftikh1@jhmi.edu (A.I.); 2Department of Dental Public Health, University of Maryland School of Dentistry, Baltimore, MD 21202, USA; dweatherspoon@umaryland.edu (D.J.W.); efleming@umaryland.edu (E.F.); 3Department of International Health, Johns Hopkins Bloomberg School of Public Health, Baltimore, MD 21205, USA; kbalsar2@jhmi.edu; 4Health Services Management, University of Maryland Global Campus, 3501 University Blvd. East, Baltimore, MD 21202, USA

**Keywords:** oral health, unmet dental need, emergency dental care, dental care access

## Abstract

Aim: Our aim was to examine the association between emergency dental care use and dental providers’ availability and ZIP code social vulnerability index. Methods: This cross-sectional observational study mapped variations in emergency dental care and analyzed their association with social vulnerability using generalized spatial two-stage least-squares to address spatial correlation. To perform spatial autoregressive modeling to examine how dental provider capacity and social vulnerability influence emergency care use across neighboring counties, accounting for ZIP Code Tabulation Area (ZCTA) spillover effects, we used secondary data from the Maryland Health Services Cost Review Commission, InfoUSA, and the US Census American Community Survey. We focused on emergency dental care usage in ZIP Code Tabulation Areas, using emergency department visits and inpatient stays per 1000 residents for dental conditions, as the dependent variable. Results: Highly vulnerable ZCTAs saw 5.9× more non-traumatic dental condition (NTDC) ED visits, 5.3× more chronic dental ED visits, 3.3× more NTDC inpatient stays, and 1.3× more chronic dental inpatient stays than less vulnerable areas. For all four measures of emergency dental care use, increased dental provider availability was associated with reductions in use of emergency dental care (ED), while higher social vulnerability was associated with increased use. For example, an increase of one dental provider per 1000 residents was associated with a reduction of 28.2 non-traumatic dental ED visits, while a high social vulnerability index was associated with an increase of 88.6 non-traumatic dental ED visits. Conclusions: There was an association among dental provider availability, social vulnerability, and the use of emergency dental care.

## 1. Introduction

Untreated dental disease is a pressing public health concern affecting millions of Americans [1,2], with implications extending beyond oral health, impacting physical function due to pain [3], learning abilities in children [4,5], and employment opportunities [6]. *The Surgeon General’s 2000 Report* highlighted the connection between oral and overall health, emphasizing the silent epidemic of oral health disparities and their effects on quality of life [7]. Oral diseases, such as dental caries and periodontal disease, have been strongly linked to chronic health conditions like diabetes and heart disease, alongside nutritional deficiencies [8,9].

Untreated dental diseases, including abscesses, are the leading cause of ED visits for non-traumatic dental conditions (NTDCs) [10,11]. Evidence indicates that regular oral evaluations, proper oral health behaviors, and evidence-based preventive care effectively minimize dental disease risk. However, emergency department visits due to untreated dental disease still pose a significant public health burden [12]. This burden is not limited to the United States. Globally, untreated dental disease affects over 3.5 billion people, making it a significant issue worldwide. Similar disparities are seen in other countries, including low- and middle-income nations, where access to preventive and restorative dental care is also limited, resulting in higher emergency dental care utilization [13].

Uninsured or publicly insured individuals frequently lack both preventive and timely restorative dental care [14]. Oral health with public coverage remains a policy concern as critical treatment is often delayed until conditions become emergencies, driving individuals to seek immediate care for preventable dental issues in hospital emergency departments [14]. The World Health Organization’s (WHO) Global Strategy on Oral Health (GOHAP) has emphasized the need to integrate oral health into primary healthcare frameworks worldwide. These global initiatives aim to reduce disparities and improve access to preventive services, a challenge seen in countries with both developed and developing healthcare systems [15].

NTDCs that result in an ED visit should be classified as “ambulatory care-sensitive condition visits, i.e., visits that are potentially avoidable through timely and effective outpatient care” [16,17]. Routine outpatient dental care can effectively minimize and manage such conditions [12].

Preventable ED visits for non-traumatic dental conditions not only impose a significant burden in terms of outcomes but also contribute substantially to healthcare costs and resource utilization [16]. Emergency department treatment for NTDCs, while costly, often lacks definitive care for the underlying issues [14]. Despite 90% of patients leaving the ED with temporary pain relief and prescriptions, many require dental follow-up to prevent exacerbation of conditions and subsequent ED return visits. Repeat ED visits signal ineffective initial treatment and potential access barriers to dental follow-up care [14]. In severe cases, some ED visits escalate to inpatient admissions, amplifying healthcare and financial burdens on patients and the healthcare system.

This study estimated the association among unmet dental needs and geographic availability of dental providers and the social vulnerability of residents in the state of Maryland. We measured unmet dental needs using ED care use. We hypothesized that communities in Maryland with more dental care providers will have increased access to preventive and restorative services, which lowers the level of unmet dental needs in the community and the use of emergency dental services.

## 2. Methods

To address the study aim, we measured unmet dental needs using ED care use. Unmet dental needs are determined by demographic and enabling factors that render some communities more vulnerable to access problems compared to others. We used the Social Vulnerability Index (SVI) to rank communities based on socioeconomic status, household composition, racially minoritized populations, composition and language, and housing type and transportation [18]. We hypothesized that more vulnerable communities in Maryland will have more unmet dental health needs and, therefore, use more emergency dental services.

This study was reviewed by the Johns Hopkins Bloomberg School of Public Health Institutional Review Board and was determined to meet the criteria for exemption under 45 CFR 46.101(b). We followed the Strengthening the Reporting of Observational Studies in Epidemiology (STROBE) reporting guidelines for cross-sectional studies.

### 2.1. Data Sources and Unit of Analysis

The data for ED visits and inpatient stays were obtained from the statewide public-use Hospital Discharge and Outpatient datasets maintained by the Maryland Health Services Cost Review Commission (MCRC, 2016, HSCRC-2016). Dental provider office data were acquired from the InfoUSA Business Layout Data (2017). Population, demographic, and socioeconomic data used to construct the SVI were pulled from the U.S. Census Bureau American Community Survey (ACS; 5 years: 2012–2016).

ZCTA (ZIP Code Tabulation Area) level was the unit of analysis. The 5-digit ZCTA was designed by the U.S. Census bureau to approximate ZIP codes. A total of 468 Maryland ZCTAs were available within the 2016 5-year American Community Survey. We excluded 54 ZCTAs with populations of less than 300 people, for a final sample of 414 ZCTAs.

### 2.2. Key Variables

We used four measures of emergency dental care: ED visits and inpatient hospital stays for NTDCs, and ED visits and inpatient hospital stays for chronic dental conditions. NTDC ED visits and stays were defined as those with primary or secondary diagnoses for preventable dental conditions, such as dental caries-related and/or periodontal conditions (See Table 1 footnote for the full list of ICD-10-CM codes). NTDC excludes diagnoses that cannot be addressed through outpatient dental care (e.g., traumatic injuries and neoplastic lesions). Chalmers’ classifications were utilized to define chronic dental conditions [19].

To measure dental provider availability, the number of dental office providers (DOPs) in a ZCTA were identified using the InfoUSA Business Layout Data by aggregating the number of dentists who practice from dental offices (NAICS Business Code = 621210). The social vulnerability index for each ZCTA was computed for all 468 Maryland ZCTAs using demographic and socioeconomic characteristics of the ZCTAs from the ACS. Using the tertile cutoff of the SVI index for the 468 ZCTAs, we categorized them into three groups: low, medium, and high SVI. We used the total population of the ZCTA to control its size. We then excluded the ZCTAs will fewer than 300 residents, reducing the sample to 414 ZCTAs. We reported the means and standard deviations for the dependent and independent variables in Table 1.

### 2.3. The Heat Map

We created a series of Maryland state maps illustrating the utilization of emergency dental services per 1000 residents in ZCTAs and compared it with the population-to-dental office provider ratio. Areas where the ratio exceeded 5000 were identified as dental provider shortage areas [20]. The color gradations in the maps are organized into deciles based on the variable. As the color transitions from yellow to red, it signifies an increase in the displayed variable’s value. Red areas indicate high utilization of emergency dental care or a shortage of dental providers in those locations.

The dental provider capacity and the social vulnerability are geographic factors that can influence emergency care use in neighboring counties. To account for this potential spillover effect of the ZCTAs, we used spatial autoregressive (SAR) modeling techniques.

### 2.4. Statistical Analysis

In our statistical analysis, we utilized generalized spatial two-stage least-squares estimator accounts for spatial lags of outcome across ZCTAs using a spatial weighting matrix which factors the adjacent/contiguous relationships of the ZCTAs. For each variable, we estimated the direct effect (i.e., ZCTA’s own direct effect), indirect effect (i.e., spillover effect from adjacent ZCTAs), and total effect (i.e., the sum of direct and indirect effects). We reported the marginal effects and standard errors.

The full model was adjusted for several demographical controls: total population, percentage of women in the population, and age distribution in percentages across categories 0–20 and 21–64 years. The level of social vulnerability was categorized into three categories with low or least vulnerable serving as the reference group for the independent variable. The study scope was restricted to ZIP code populations greater than or equal to 300. To ensure robustness, the analysis employed the Wald Test of Spatial Terms (*p* < 0.05) to validate the significance of spatial components in our models. Additionally, chronic dental conditions were defined based on established criteria by Chalmers, 2017 [19].

SAS software, Version 9.4 64-bit (Copyright© 2019, SAS Institute Inc., Cary, NC, USA) and STATA [Version 18] (StataCorp, College Station, TX, USA) were employed to analyze the data and create the analytical dataset containing the final outcome variables. These variables were then superimposed onto the maps. The maps were generated using SAS/GRAPH^®^ X procedures, with the presentation of the Maryland ZCTAs based on the Census 2010 TIGER/Line^®^ Shape Files.

## 3. Results

### 3.1. The Geographic Distribution of Hospital-Based Dental Services

Distribution of emergency department visits and inpatient stays: Figure 1 and Figure 2 display the geographic distribution of emergency dental care use. Figure 1 shows the number of ED visits for NTDCs per 1000 people in the ZCTAs.

Figure 2 displays the number of inpatient stays per NTDCs per 1000 individuals in the ZCTAs. Concerning ED utilization, multiple ZCTAs on the Eastern Shore and many in Baltimore City, along with select areas in Allegany, Charles, Calvert, St. Mary’s, and Garrett counties, exhibit the highest deciles of ED visits. In contrast, the localities with the most inpatient stays include ZCTAs in Baltimore City and its surrounding suburbs. Rural counties also display pockets of elevated inpatient stays. Moreover, ZCTAs in the top 3 deciles appear sporadically dispersed with minimal gradation, unlike the more uniform distributions observed for ED visits. The counties in the Washington D.C. metro area (Montgomery, Prince George’s, and Howard counties) had the lowest rates of emergency dental care use.

Distribution of dental office providers: Figure 3 illustrates the population-to-DOP ratios in each ZCTA. These maps reveal clusters of ZCTAs with limited DOPs and how their supply changes when factoring in geographic access to DOPs within a 3- or 5-mile radius of the ZCTA. The red areas on the maps indicate ZCTAs where the population-to-DOP ratio exceeds the Health Professional Shortage Area (HRSA) thresholds [20]. There are 116 ZCTAs with a DOP ratio greater than 5000:1 or with zero DOPs. Most of these ZCTAs are in the rural Maryland counties. Among the rural counties, the southern counties (St. Mary’s Frederick and Charles), the Eastern Shore counties (Wicomico, Dorchester, Queen Anne’s, and Talbot) and Allegany County in Western Maryland have the most ZCTAs with a dental shortage. There are pockets of shortage areas in northern ZCTA reaches of the other urban Maryland counties (Anne Arundel, Baltimore, Carroll, Harford, and Frederick).

### 3.2. The Association Between Provider Availability and Use of Hospital-Based Dental Services

In our study, we focused on analyzing emergency dental care utilization across various social vulnerability index (SVI) categories. The data were restricted to ZIP codes with populations greater than or equal to 300, excluding six specific ZIP codes that comprised universities, prisons, and other non-residential areas. These exclusions ensured a more accurate reflection of typical residential communities. The analysis involved population-weighted metrics, disregarding the total population count, and the SVI was divided into low, medium, and high categories based on tertile cutoffs, with low representing the least vulnerable and high indicating the most vulnerable groups.

We included a range of ICD-10-CM codes encompassing various dental conditions as defined by Chalmers in 2017. These included A690, K023, K0251, K0252, K0253, K0261, K0262, K0263, K027, K029, K0389, K040, K0401, K0402, K041, K042, K044, K045, K046, K047, K048, K0490, K0499, K0500, K0510, K0520, K0521, K05211, K05219, K0530, K0531, K05319, K055, K056, K08129, K08139, K083, K08439, K08530, K08531, K08539, K0859, K088, K089, K122, K1370, M273, M2751, M2752, M2753, and M2759. This comprehensive approach allowed us to explore the intricate relationships between social vulnerability and the need for emergency dental services in distinct demographic groups.

Table 1 displays the descriptive statistics for the multivariable analysis. We reported the mean number of ED visits and inpatient stays per ZCTA for NTDCs and chronic dental conditions. High socially vulnerable ZCTAs had 5.9 times more NTDC ED visits and 5.3 times more chronic dental-related ED visits compared to low socially vulnerable ZCTAs. High socially vulnerable ZCTAs had 3.3 times more NTDC inpatient stays and 1.3 times more chronic dental-related inpatient stays compared to low socially vulnerable ZCTAs.

In Table 2, we report the regression results for the NTDCs and chronic dental conditions. We found that an increase in DOPs per 1000 residents was associated with reductions in ED visits for NTDCs and chronic dental-related conditions. An increase of one DOP per 1000 residents was associated with a total reduction of 28.5 ED visits due to NTDCs and 50.0 ED visits for chronic dental conditions. High social vulnerability was associated with large increases in ED visits for NTDCs and chronic dental-related conditions. ZCTAs with high social vulnerability had 105.5 more NTDC ED visits and 134.4 more chronic dental-related ED visits compared to ZCTAs with low social vulnerability. About 48% of the total effects were attributable to spillover effects from neighboring ZCTAs. We found a similar association for hospital inpatient stays. An increase of one DOP per 1000 residents was associated with a reduction of 0.27 inpatient NTDC stays and 0.32 inpatient chronic dental-related conditions, while ZCTAs with high social vulnerability had 0.81 more inpatient NTDC stays and 0.59 more inpatient chronic dental-related conditions compared to ZCTAs with low social vulnerability.

## 4. Discussion

We analyzed the spatial distribution of dental provider shortages and emergency dental service utilization for preventable NTDCs. The findings reveal an association among provider availability, area-level social vulnerability, and patient utilization of emergency dental services. Areas with provider shortages often coincide with higher emergency service utilization in socially vulnerable regions, impacting patients in adjacent ZCTAs. While increasing dental providers may enhance preventive care and reduce emergencies [21], our results suggest that addressing social vulnerability-related barriers to care is crucial, as simply expanding geographic provider access may not fully mitigate the need for emergency services. Policymakers should consider comprehensive strategies to improve dental care access and address social vulnerability barriers.

ED utilization may not be solely due to geographic accessibility but also to other barriers hindering regular dental care access. Previous research has suggested that these barriers may include limited ambulatory care hours in rural settings, decreased perception of needs among rural residents, and financial barriers caused by fewer insurance benefits among privately insured rural residents and difficulty in enrolling for Medicaid [22,23,24,25]. This study offers a nuanced view that suggests multiple factors influence care access beyond geography. Social vulnerability introduces competing obstacles affecting regular care versus ED visits. While increased distances to rural EDs correlate with office care utilization, it complicates assumptions on ED visits as a result of dental office inaccessibility [26].

Allegany and Calvert counties are in the top quartile for per capita ED dental service usage, while most areas in Frederick County are in the top ED visit quartiles. Community dental initiatives, like ED diversion programs and university health center partnerships, have been introduced to lessen emergency dental service reliance and lower costs in these regions—Allegany County, Calvert County, and, more recently, Frederick County [27,28,29,30]. Assessing these programs’ effectiveness in reducing avoidable dental ED visits and their broader impact on oral health access at the community level is a critical area for further research.

The global oral health agenda is based on six guiding principles outlined in the Global Strategy on Oral Health, which also inform the development of the WHO GOHAP for 2023–2030. These principles, designed to aid implementation in member states, include adopting a public health approach, integrating oral health into primary healthcare, employing innovative workforce models, providing people-centered care, tailoring interventions throughout the lifespan, and leveraging digital technologies to enhance oral health outcomes [15]. Our findings align with global patterns observed in countries such as the UK [31], Canada [32], and Australia [33], where higher social vulnerability is also linked to increased emergency dental care utilization. This consistency across different healthcare systems suggests that interventions like improving provider availability and reducing socioeconomic barriers may have broader applicability and could be adapted internationally. Aligning with the WHO GOHAP [15], these results emphasize the need for comprehensive global strategies to integrate oral health into primary healthcare frameworks.

In the direction of the WHO strategic principles, in 2023, the state of Maryland began providing dental benefits for adults enrolled in Medicaid [34]. While it is too early to assess the impact of this policy change on access to oral healthcare, the data analyzed in this study (prior to the enactment of the policy) can serve as valuable baseline data to help assess the impact of the new policy through future analyses. Moreover, with the expansion of dental benefits for adults with Medicaid insurance, future studies may want to access the degree to which having dental insurance precludes the use of care in ED settings. Programs such as educational campaigns, community workshops, media outreach, and online platforms and apps could focus on strategies to increase awareness of oral health to overall health, delivering significant research findings to a potential range of stakeholders and policymakers engaged in bringing systems change in oral health programs.

### Limitations

Several aspects of this study require clarification. The dental provider counts obtained from InfoUSA Business Layout may underestimate dentists in low-income and rural areas, as some providers work part-time at Federally Qualified Health Centers FQHCs and safety net clinics, activities not always recorded in InfoUSA data. Dental practices within larger medical institutions were not categorized as dental offices, potentially impacting the counts. Coding miscues in dental-related ED visits based on ICD-10-CM codes could lead to patient misclassification, affecting data accuracy. The exclusion of 54 Maryland ZCTAs may impact spillover effects on the SAR models [10,35,36,37]. Generalizability beyond Maryland may be limited. Emergency dental care use could be both a cause and a consequence of social vulnerability. Similarly, social vulnerability may lead to limited access to dental care, and conversely, restricted dental care access could exacerbate social vulnerability. Nevertheless, the study has several strengths; for example, it is the first to examine the association between the Social Vulnerability Index (SVI) and dental care in Maryland. Additionally, we included a range of ICD-10-CM codes covering various dental conditions as defined by Chalmers in 2017. This comprehensive approach allowed us to explore the intricate relationships between social vulnerability and the need for emergency dental services across different demographic groups.

## 5. Conclusions

The findings indicate that provider shortages in socially vulnerable areas lead to higher emergency dental service utilization, impacting nearby regions. Addressing social vulnerability is crucial to reducing emergency dependency, even with increased provider availability. Expanding the availability of dental providers can help reduce the reliance on emergency department and inpatient hospital services for preventable non-traumatic dental conditions. The study highlights how social vulnerability and systemic barriers impact healthcare access. Increasing the accessible provider workforce is vital to reducing emergency dental care and untreated conditions. Developing policies for Medicaid dental coverage is essential.

## Figures and Tables

**Figure 1 healthcare-12-02191-f001:**
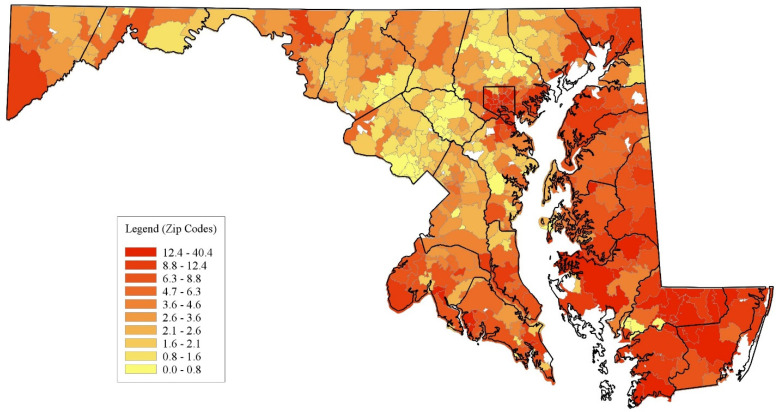
Geographic distribution of non-traumatic dental visits per 1000 in Maryland. Source: Authors’ calculations based on Maryland Hospital Utilization Data (2016).

**Figure 2 healthcare-12-02191-f002:**
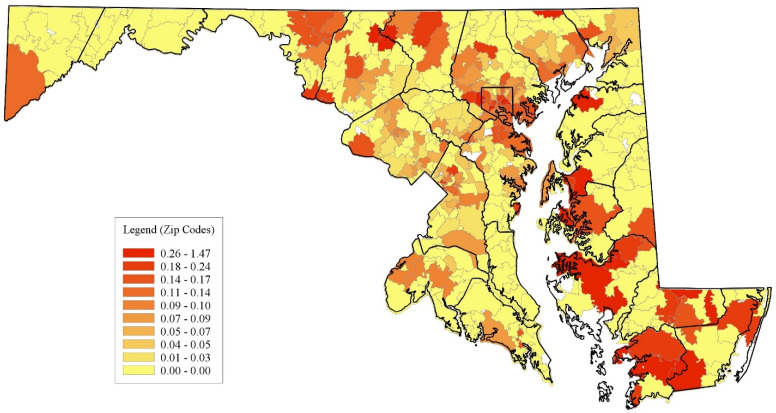
Geographic distribution of non-traumatic dental stays per 1000 in Maryland. Source: Authors’ calculations based on Maryland Hospital Utilization Data (2016).

**Figure 3 healthcare-12-02191-f003:**
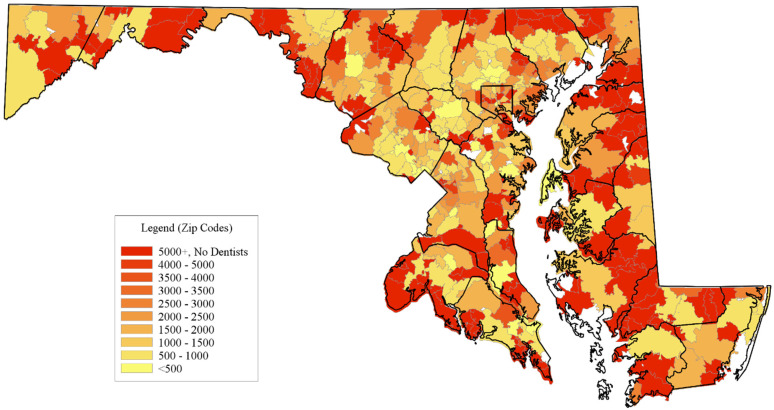
Geographic distribution of dental providers using the population to dental office provider ratio in Maryland. Source: Authors’ calculations based on Dental Provider Data from InfoUSA Business Layout Data (2017) and population data from the U.S. Census Bureau American Community Survey (ACS; 5 years: 2012–2016).

**Table 1 healthcare-12-02191-t001:** Mean number of ED visits and inpatient stays, dental office providers per 1000, and population demographic and socioeconomic factors by overall social vulnerability using the Social Vulnerability Index (SVI) in FY 2016.

Variable	All ZIP Codes (N = 414)	Low SVI (N = 118)	Medium SVI (N = 145)	High SVI (N = 151)	*p*-Value
Outcome [Mean (SD)]					
Non-Traumatic Dental-Related ED Visits	186.1 (226.2)	46.6 (68.9)	61.7 (49.0)	274.4 (255.4)	<0.001
Chronic Dental-Related ^+^ ED Visits	258.5 (298.7)	70.5 (95.9)	93.1 (69.1)	376.5 (335.4)	<0.001
Non-Traumatic Dental-Related Inpatient Stays	2.4 (2.73)	1.0 (1.9)	1.2 (1.3)	3.3 (3.0)	<0.001
Chronic Dental-Related ^+^ Inpatient Stays	3.9 (3.65)	2.5 (3.2)	2.3 (2.2)	4.9 (3.9)	<0.001
Dental Capacity					
Dental Office Providers per 1000	0.9 (0.8)	0.7 (0.6)	1.1 (0.9)	0.9 (0.8)	<0.001
Control Variables					
Total Population (ACS)	14,333 (15,936)	5802 (9443.0)	11,944 (12,599)	23,296 (182,66)	<0.001
Percent Men, (%)	48.4 (2.5)	49.8 (3.2)	48.4 (2.0)	48.1 (2.5)	0.234
Percent Women, (%)	51.62 (2.5)	50.2 (3.2)	51.6 (2.0)	51.9 (2.5)	0.234
Percent Age 0–20, (%)	26.5 (3.7)	27.6 (4.3)	27.0 (3.6)	26.1 (3.6)	0.984
Percent Age 21–64, (%)	59.7 (4.0)	57.7 (3.8)	58.7 (3.8)	60.5 (4.0)	<0.001
Percent Age 65+, (%)	13.8 (4.5)	14.7 (4.5)	14.3 (5.0)	13.3 (4.2)	0.005

Sources: The 2016 HSCRC Outpatient/Inpatient; 2017 InfoUSA; ACS 2016 5 years (2012–2016). Notes: Restricted to ZIP code populations greater than or equal to 300; dropped additional 6 ZIP codes. Included university, prison, and other non-residential ZIP codes; population-weighted (excluding total population); low, medium, and high categories for the SVI were generated based on tertile cutoffs of the SVI; low—least vulnerable; high—most vulnerable; Analysis of Variance (ANOVA) test was used to test differences between SVI groups. ICD-10-CM Codes: A690, K023, K0251, K0252, K0253, K0261, K0262, K0263, K027, K029, K0389, K040, K0401, K0402, K041, K042, K044, K045, K046, K047, K048, K0490, K0499, K0500, K0510, K0520, K0521, K05211, K05219, K0530, K0531, K05319, K055, K056, K08129, K08139, K083, K08439, K08530, K08531, K08539, K0859, K088, K089, K122, K1370, M273, M2751, M2752, M2753, and M2759. + Dental condition definition (Chalmers 2017 [19]).

**Table 2 healthcare-12-02191-t002:** Association between emergency dental care and dental provider availability and Zip Code Social Vulnerability Index (SVI) in Maryland, 2016.

	Direct dy/dx (SE)	Indirect dy/dx (SE)	Total dy/dx (SE)
Non-Traumatic Dental-Related ED Visits			
Dental Office Providers per Capita in ZIP	−20.219 (6.367) **	−18.290 (7.027) **	−38.510 (12.713) **
SVI: Medium	−14.583 (12.683)	−13.192 (12.147)	−27.774 (24.635)
SVI: High—Most Vulnerable	55.381 (13.950) ***	50.098 (15.598) **	105.479 (27.134) ***
Chronic Dental-Related ^+^ ED Visits			
Dental Office Providers per Capita in ZIP	−26.930 (8.175) **	−23.107 (8.541) **	−50.037 (15.893) **
SVI: Medium	−17.908 (16.271)	−15.366 (14.631)	−33.275 (30.698)
SVI: High—Most Vulnerable	72.315 (17.894) ***	62.048 (18.771) **	134.363 (33.833) ***
Non-Traumatic Dental-Related Inpatient Stays			
Dental Office Providers per Capita in ZIP	−0.194 (0.084) *	−0.077 (0.041)	−0.270 (0.119) *
SVI: Medium	−0.022 (0.167)	−0.009 (0.067)	−0.031 (0.234)
SVI: High—Most Vulnerable	0.578 (0.184) **	0.229 (0.099) *	0.807 (0.258) **
Chronic Dental-Related ^+^ Inpatient Stays			
Dental Office Providers per Capita in ZIP	−0.251 (0.112) *	−0.067 (0.040)	−0.318 (0.145) *
SVI: Medium	−0.050 (0.222)	−0.013 (0.059)	−0.063 (0.281)
SVI: High—Most Vulnerable	0.465 (0.242)	0.124 (0.078)	0.589 (0.308)

Sources: 2016 HSCRC Outpatient/Inpatient; 2017 InfoUSA; ACS 2016 5 years (2012–2016). Notes: * *p* < 0.05; ** *p* < 0.01, and *** *p* < 0.001. Generalized spatial two-stage least-squares autoregressive models were run accounting for a spatially lagged dependent variable. Spatially weighted by the contiguous distance of other ZCTAs; full model controlled by total population, %women, %age 0–20, and %age 21–64; independent variable reference: SVI low—least vulnerable; restricted to ZIP code populations greater than or equal to 300; dropped additional 6 ZIP codes. Included university and prison ZIP codes. ^+^ Chronic dental condition definition [19].

## Data Availability

The data are not publicly available and are subject to a data usage agreement between the Hilltop Institute and Johns Hopkins University.

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
