# Peer review of "Spatial Patterns of Emergency Dental Care Utilization: Assessing the Influence of Social Vulnerability and Dental Provider Availability"

_healthcare, 2024, doi:10.3390/healthcare12212191_

Round 1
Reviewer 1 Report
Comments and Suggestions for Authors
Dear Authors,
I have reviewed your manuscript titled "Emergency Dental Care Utilization: Assessing the Influence of Social Vulnerability and Dental Provider Availability." Overall, I found the article to be well-written and thoroughly documented.
The introduction provides a comprehensive overview of the study’s context and clearly outlines the importance of examining the influence of social vulnerability and dental provider availability on emergency dental care utilization.
The choice of data sources and analytical methods are appropriate and well-justified, and they align with the study's objectives.
The results are presented clearly and systematically. The use of tables and figures enhances the reader’s understanding of the data.
The discussion is well-organized and provides a thoughtful interpretation of the results in the context of existing literature.
The conclusions are concise and appropriately reflect the study's findings. I recommend that the conclusion section be rewritten to include specific outcomes and results from the study, as I noticed this information is lacking. Currently, the conclusion focuses too much on offering suggestions for reducing the reliance on emergency departments, and it would benefit from a stronger emphasis on the actual findings of the research.
I thank the authors for their hard work in analyzing all the data and compiling this informative manuscript.
Best regards
Author Response
Reviewer #1
I have reviewed your manuscript titled "Emergency Dental Care Utilization: Assessing the Influence of Social Vulnerability and Dental Provider Availability." Overall, I found the article to be well-written and thoroughly documented.
Comment#1: The introduction provides a comprehensive overview of the study’s context and clearly outlines the importance of examining the influence of social vulnerability and dental provider availability on emergency dental care utilization. The choice of data sources and analytical methods are appropriate and well-justified, and they align with the study's objectives. The results are presented clearly and systematically. The use of tables and figures enhances the reader’s understanding of the data. The discussion is well organized and provides a thoughtful interpretation of the results in the context of existing literature. The conclusions are concise and appropriately reflect the study's findings.
Response#1: Dear reviewer, thank you very much for your kind words. We truly appreciate your feedback.
Comment#2: I recommend that the conclusion section be rewritten to include specific outcomes and results from the study, as I noticed this information is lacking. Currently, the conclusion focuses too much on offering suggestions for reducing the reliance on emergency departments, and it would benefit from a stronger emphasis on the actual findings of the research.
I thank the authors for their hard work in analyzing all the data and compiling this informative manuscript.
Response#2: We revised the conclusion to address the reviewer’s comment.
Dear Reviewer, we thank you once again for your most valuable comments and appreciate having had this wonderful opportunity to learn from you. We hope that our responses have addressed your comments effectively.
Reviewer 2 Report
Comments and Suggestions for Authors
The manuscript should be improved;
Remove academic references (PhD, MS etc) from authors. Affiliations only refers to the centre or institution which belongs authors, not their work (Dean, assistant etc should be removed).
1- ABSTRACT: include statistical analysis in methods, and data in results. Remove the last phrase "personal opinion" from conclusion.
2- INTRODUCTION: too short. It should be augmented. Add global prevalence, not only American. Re-write the last paragraph following antecedents, justification, hypothesis and objective. Not discuss the method in this section.
3-METHODS: add study design, add STROBE reference, subjects authorised data investigation use?, remove results tables from methods.
4- RESULTS: add result tables here, improve figure quality.
5- DISCUSSION: add methods' discussion, is there similar studies?, discuss with other countries not only USA, add clinical implication.
6- CONCLUSION: remove references and personal opinion. Conclusion should answer the study objective.
Author Response
Reviewer #2
Comment#1: Remove academic references (PhD, MS etc) from authors. Affiliations only refers to the center or institution which belongs authors, not their work (Dean, assistant etc. should be removed).
Response#1: Thank you for the comments, we edited the authors affiliations to address the reviewer’s comment.
Comment#2: 1- ABSTRACT: include statistical analysis in methods, and data in results. Remove the last phrase "personal opinion" from conclusion.
Response#2: We have modified the abstract to address this comment, thank you.
Comment#3: 2- INTRODUCTION: too short. It should be augmented. Add global prevalence, not only American. Re-write the last paragraph following antecedents, justification, hypothesis and objective. Not discuss the method in this section.
Response#3: We have modified the introduction to address this comment, thank you.
Comment#4: 3-METHODS: add study design, add STROBE reference, subjects authorized data investigation use? remove results tables from methods.
Response#4: We agree with the reviewer, we have modified the text to address the comments.
Comment#5: 4- RESULTS: add result tables here, improve figure quality.
Response#5: Thank you for the comment, we add the PDF version of the figures for the publication (In case the paper has been accepted for the publication).
Comment#6: 5- DISCUSSION: add methods' discussion, is there similar studies? discuss with other countries not only USA, add clinical implication.
Response#6: Thank you for your comments. However, for this study, we have focused exclusively on the US. As a result, we have not included any literature from outside the US.
Comment#7: 6- CONCLUSION: remove references and personal opinion. Conclusion should answer the study objective.
Response#7: Thank you for the comment. Considering the comments from both the first and second reviewers, we have modified the conclusion accordingly.
Dear Reviewer, we thank you once again for your most valuable comments and appreciate having had this wonderful opportunity to learn from you. We hope that our responses have addressed your comments effectively.
Reviewer 3 Report
Comments and Suggestions for Authors
Thank You for the opportunity to review.
A well-written manuscript, which highlights the need for effective policy to increase the dental workforce, and reduce the burden on ER.
1. Authors please check for gross grammatical errors, there are a couple of them in the text and conclusion.
Comments on the Quality of English LanguageGrammatical check required.
Author Response
Reviewer #3
A well-written manuscript, which highlights the need for effective policy to increase the dental workforce, and reduce the burden on ER.
Comment#1: 1. Authors please check for gross grammatical errors, there are a couple of them in the text and conclusion.
Respond #1: Dear Reviewer, we thank you once again for your most valuable comment and appreciate having had this wonderful opportunity to learn from you. We hope that our responses have addressed your comments effectively.
Reviewer 4 Report
Comments and Suggestions for Authors
Dear authors,
I have some considerations about your manuscript.
First of all, in the title, it would be better if you add the type of study.
In relation to introduction:
The objective includes a part of the methodology of the study, please you should write it better and clearer.
In relation to material and methods:
Was the sample size calculated?
You should explain better the inclusion criteria of the sample.
How many people collected the data?
Please, it should be interesting to add a material part. It does not appear in the text.
Did the present study was approved by an ethics committee?
In relation with table 1, you can explain de variables in this part, but the table is part of the results.
The same with table 2.
In relation to results:
In relation with the figures, it could be interesting to create tables where data, data means, standard deviations and, if it´s necessary statistical significance appeared.
In relation to discussion:
Discussion is misspelled, it is in the singular, not the plural.
It is too short, please expand with similar studies that have been carried out in other populations.
It would be interesting to add strengths of the study.

Author Response
Reviewer #4
I have some considerations about your manuscript.
Comment#1: First of all, in the title, it would be better if you add the type of study.
Respond #1: Thank you for the comment, we modified the title to address the reviewer’s comment:
This is the new title:
Spatial Patterns of Emergency Dental Care Utilization: Assessing the Influence of Social Vulnerability and Dental Provider Availability
Comment#2: In relation to introduction: The objective includes a part of the methodology of the study, please you should write it better and clearer.
Respond #2: We agree with the reviewer's comments and have accordingly modified both the introduction and methods section to address these comments.
Comment#3: In relation to material and methods:
Comment#3a: Was the sample size calculated?
Comment#3b: You should explain better the inclusion criteria of the sample.
Comment#3c: How many people collected the data? Please, it should be interesting to add a material part. It does not appear in the text.
Response Comment#3a, b, and c: We started off with a total of 468 Maryland ZCTAs where we had categorized our SVI Index into three groups of low, medium and high. Restricting the sample to fewer than 300 residents, we ended up with a final sample of 414 ZCTAs.
Comment#3e: Did the present study was approved by an ethics committee?
Response: Yes. This study was reviewed by the Johns Hopkins Bloomberg School of Public Health Institutional Review Board and was determined to meet the criteria for exemption under 45 CFR 46.101(b). We followed Strengthening the Reporting of Observational Studies in Epidemiology (STROBE) reporting guidelines for cross sectional studies.
Comment#3f: In relation with table 1, you can explain dependent variables in this part, but the table is part of the results. The same with table 2.
Response: We modified the text to address this comment.
Comment#4: In relation to results:
In relation with the figures, it could be interesting to create tables where data, data means, standard deviations and, if it´s necessary statistical significance appeared.
Response#4: Thank you for the comment. We have updated Table 1 to include all variables used in Figures 1, 2, and 3, categorized according to SVI. Please let us know if you have any further suggestions.
Comment#5: In relation to discussion:
Discussion is misspelled, it is in the singular, not the plural.
It is too short, please expand with similar studies that have been carried out in other populations.
It would be interesting to add strengths of the study.
Response#5, thank you we edited the text, added the strengths of the study, to address these comments.
Dear Reviewer, we thank you once again for your most valuable comments and appreciate having had this wonderful opportunity to learn from you. We hope that our responses have addressed your comments effectively.
Reviewer 5 Report
Comments and Suggestions for Authors
The manuscript entitled “Emergency Dental Care Utilization: Assessing the Influence of Social Vulnerability and Dental Provider Availability” is dealing with very “sensitive” public healthcare problem of dental care accessibility. Some issues need to be addressed before publishing:
In methodology, the authors report that a spatial weighting matrix which factors the adjacent/contiguous relationships of the ZCTA was used, implying that emergency dental care use is influenced by directly adjacent areas. Are there any data from areas beyond?
Please, address the limitations of the modelling used, e.g., the variables influence each other and the direction is unclear: emergency dental care use could both result from and contribute to social vulnerability, or social vulnerability leads to limited dental care access, and limited dental care access could increase social vulnerability.
Please, could you interpret the direct, indirect, and total effects in more detail.
Are there any data from other states beside Maryland?
How do you comment WHO Global Strategy and Action Plan on Oral Health 2023–2030?
Could the major milestones defined by WHO be achieved by 2030?
Author Response
Reviewer #5
Comment#1:The manuscript entitled “Emergency Dental Care Utilization: Assessing the Influence of Social Vulnerability and Dental Provider Availability” is dealing with very “sensitive” public healthcare problem of dental care accessibility. Some issues need to be addressed before publishing: In methodology, the authors report that a spatial weighting matrix which factors the adjacent/contiguous relationships of the ZCTA was used, implying that emergency dental care use is influenced by directly adjacent areas. Are there any data from areas beyond?
Response #1: We did not look at the influence of a zip code that is not contiguous. That would require different types of spatial analysis.
Comment#2: Please, address the limitations of the modelling used, e.g., the variables influence each other, and the direction is unclear: emergency dental care use could both result from and contribute to social vulnerability, or social vulnerability leads to limited dental care access, and limited dental care access could increase social vulnerability.
Response #2: We agree with the reviewer, we added a sentence to the limitation section to address this comment.
Comment#3: Please, could you interpret the direct, indirect, and total effects in more detail.
Are there any data from other states besides Maryland?
Thank you for the comment. For this study, we have utilized data from Maryland. Including data from other states would require access to additional datasets, which, based on our experience, involves a time-consuming and costly process that can exceed a year.
Comment#4: How do you comment WHO Global Strategy and Action Plan on Oral Health 2023–2030? Could the major milestones defined by WHO be achieved by 2030?
Response#4: In 2023, the state of Maryland began providing dental benefits for adults enrolled in Medicaid, to evaluate the effect of this policy, more study needs to be done.
Dear Reviewer, we thank you once again for your most valuable comments and appreciate having had this wonderful opportunity to learn from you. We hope that our responses have addressed your comments effectively.
Round 2
Reviewer 2 Report
Comments and Suggestions for Authors
- Hypothesis must be in introduction prior to objective, not in MM section
- Remove ethical aceptance and STROBE guideline from Statistical section, move it to the general methods section (before 2.1 section)
- Without adding international information and discussion this manuscript can't be published in an international journal because results must be discussed with international authors and results, making it extrapolable to other populations.
- Resume conclusions, too long.
Author Response
Comment#1: Hypothesis must be in introduction prior to objective, not in MM section
Thank you for the comment; we have moved the hypotheses to the introduction.
Comment#2: Remove ethical acceptance and STROBE guideline from Statistical section, move it to the general methods section (before 2.1 section)
Thank you for the comment; we have moved the related text before section 2.1.
Comment#3: Without adding international information and discussion this manuscript can't be published in an international journal because results must be discussed with international authors and results, making it extrapolable to other populations.
Thank you for the comment; we have revised the introduction and discussion sections to address your feedback.
Comment#4: Resume conclusions, too long.
Thank you; we have condensed it.
Dear Reviewer, we thank you once again for your most valuable comments and appreciate having had this wonderful opportunity to learn from you. We hope that our responses have addressed your comments effectively.
Reviewer 4 Report
Comments and Suggestions for Authors
Dear authors
I have not more considerations
Author Response
Dear Reviewer, we thank you once again for your most valuable comments in the first round and appreciate having had this wonderful opportunity to learn from you.